# Enhancing the Therapeutic Potential of *CCL*2-Overexpressing Mesenchymal Stem Cells in Acute Stroke

**DOI:** 10.3390/ijms21207795

**Published:** 2020-10-21

**Authors:** Sanghun Lee, Ok Joon Kim, Kee Ook Lee, Hyeju Jung, Seung-Hun Oh, Nam Keun Kim

**Affiliations:** 1Department of Neurology, CHA Bundang Medical Center, CHA University, Seongnam 13496, Korea; shlee2251@daum.net (S.L.); niceiatros@cha.ac.kr (K.O.L.); heaju0420@daum.net (H.J.); ohsh72@chamc.co.kr (S.-H.O.); 2Institute for Clinical Research, CHA Bundang Medical Center, CHA University, Seongnam 13488, Korea; nkkim@chamc.co.kr

**Keywords:** angiogenesis, CCL2, CCR2, mesenchymal stem cell, neurogenesis, stroke

## Abstract

Although intravenous administration of mesenchymal stem cells (MSCs) is effective for experimental stroke, low engraftment and the limited functional capacity of transplanted cells are critical hurdles for clinical applications. C–C motif chemokine ligand 2 (CCL2) is associated with neurological repair after stroke and delivery of various cells into the brain via CCL2/CCR2 (CCL2 receptor) interaction. In this study, after CCL2-overexpressing human umbilical cord-derived MSCs (hUC-MSCs) were intravenously transplanted with mannitol in rats with middle cerebral arterial occlusion, we compared the differences between four different treatment groups: mannitol + CCL2-overexpressing hUC-MSCs (CCL2-MSC), mannitol + naïve hUC-MSCs (M-MSC), mannitol only, and control. At four-weeks post-transplantation, the CCL2-MSC group showed significantly better functional recovery and smaller stroke volume relative to the other groups. Additionally, we observed upregulated levels of CCR2 in acute ischemic brain and the increase of migrated stem cells into these areas in the CCL2-MSC group relative to the M-MSC. Moreover, the CCL2-MSC group displayed increased angiogenesis and endogenous neurogenesis, decreased neuro-inflammation but with increased healing-process inflammatory cells relative to other groups. These findings indicated that CCL2-overexpressing hUC-MSCs showed better functional recovery relative to naïve hUC-MSCs according to the increased migration of these cells into brain areas of higher CCR2 expression, thereby promoting subsequent endogenous brain repair.

## 1. Introduction

Stroke is a leading cause of death and disability worldwide and represents a significant public health concern along with a large socio-economic burden on healthcare systems [1]. In South Korea, one in 40 adults are stroke patients, and 232 people in 100,000 experience a stroke event annually. Stroke mortality is gradually declining but remains high at 30 deaths per 100,000 individuals [2].

Currently, recombinant tissue plasminogen activator (r-tPA) is the only therapeutic agent approved by the Food and Drug Administration to treat patients with acute ischemic stroke. However, r-tPA involves highly restrictive inclusion criteria, and the clinical benefits only manifest upon administration within 4.5 h of stroke onset. Therefore, r-tPA has a narrow time window and a small proportion of eligible patients. Furthermore, serious neurological sequelae after using r-tPA remain in many cases. Moreover, to date, there remains no definitive treatment to regenerate damaged brain cells in order to restore neurological deficits following stroke.

Recently, stem cell therapies using different cell types, such as mesenchymal stem cells (MSCs), bone marrow mononuclear cells, and neural stem cells, have emerged as a promising regenerative treatment for stroke survivors with residual deficits [3,4,5]. Among these, MSCs as adult stem cell haves been extensively investigated experimentally due to their easy isolation and amplification, multipotent differentiation capacity, low immunogenicity, and their potential for paracrine and immunomodulatory functions [6,7]. The effectiveness of MSC-based treatment in stroke animal models has been demonstrated [5,8]; however, their efficacy in clinical trials involving stroke patients is controversial [9]. To address this limitation, genetically engineered MSCs for functional augmentation of various factors, such as brain-derived neurotrophic factor or glial-derived neurotrophic factor, have been investigated [6,7,10].

Chemokines (chemoattractant cytokines) are a superfamily of structurally related pro-inflammatory peptides that mediate cell-specific, directed migration of leukocytes into sites of inflammation [10]. C–C motif chemokine ligand 2 (CCL2) belongs to the C–C chemokine family, and its effects are mediated through C–C motif receptor 2 (CCR2). A recent report indicated that CCL2 is associated with neurological repair after stroke and delivery of various cells into the brain via CCL2/CCR2 interaction [11,12]. Therefore, in the present study, we evaluated the potential therapeutic effect and safety of CCL2-overexpressing MSCs in a highly relevant animal stroke model.

## 2. Results

### 2.1. Confirmation of CCL2 Overexpression in Human Umbilical Cord-Derived Mesenchymal Stem Cell (hUC-MSCs)

To confirm transfection of the CCL2-overexpression plasmid into MSCs, we verified GFP expression by fluorescence microscopy (Figure 1A) and upregulated CCL2 mRNA levels relative to controls (by ~1352-fold; *p* < 0.002) (Figure 1B). Additionally, western blot analysis indicated upregulated CCL2 levels in CCL2-overexpressing hUC-MSCs (Figure 1C, Appendix A). These findings confirmed the efficacy of Lipofectamine transfection for sustained MSC viability and gene delivery [13].

### 2.2. Improvement of Functional Deficits in Middle Cerebral Artery Occlusion (MCAo) Model Rats after Transplantation of CCL2-Overexpressing hUC-MSCs

To investigate the therapeutic efficacy of CCL2-overexpressing hUC-MSCs, we performed behavioral testing of each treatment group seven times during the 28-day study period. The results of the modified neurological severity score (mNSS) test revealed that the MAN group showed no significant behavioral changes relative to the control group, whereas the M-MSC group showed slight functional improvement in all three behavioral tests (*p* > 0.05). However, compared with the other groups, the CCL2-MSC group showed significant functional improvement starting from 1-day post-transplantation that remained robust after the second transplant and was sustained to the last day of the test (*p* < 0.01) (Figure 2A). Similarly, in the stepping test, the CCL2-MSC group showed gradual improvement of forelimb dysfunction from one-day post-transplantation relative to the other groups, with this recovery maintained for up to 28 days (*p* < 0.01) (Figure 2B). Moreover, for the foot-fault test, the CCL2-MSC group showed significantly reduced step errors from 7- to 21-days post-transplantation relative to the other groups (*p* < 0.05) (Figure 2C).

### 2.3. Reduced Infarct Size Following Transplantation of CCL2-Overexpressing hUC-MSCs

At 28 days after MCAo induction, we observed no significant change in the infarct size between the control and MAN groups but a significant decrease in infarct size in the M-MSC (control vs. M-MSC: 31.47% ± 0.62 vs. 26.67 ± 1.00; *p* < 0.001) and CCL2-MSC groups (control vs. CCL2-MSC: 31.47% ± 0.62 vs. 23.48 ± 1.55; *p* < 0.001) according to TTC staining (Figure 3A). Additionally, Cresyl Violet staining revealed similar results, with no significant difference in infarct size between control and MAN groups but significantly reduced sizes in the M-MSC (control vs. M-MSC: 23.3% ± 3.75 vs. 19.59 ± 1.36; *p* < 0.05) and CCL2-MSC groups (control vs. CCL2-MSC: 23.30% ± 3.75 vs. 15.80 ± 1.53; *p* < 0.001) (Figure 3B). Furthermore, both TTC (M-MSC vs. CCL2-MSC: 26.67% ± 1.00 vs. 23.48 ± 1.55; *p* = 0.004) and Cresyl Violet (M-MSC vs. CCL2-MSC: 19.59% ± 1.36 vs. 15.80 ± 1.53, *p* < 0.05) staining showed that the CCL2-MSC group exhibited more significant reductions in infarct size relative to the M-MSC group.

### 2.4. CCL2 and CCR2 Levels are Elevated in the Brain During the Acute Ischemic Phase

We found that CCL2 levels were significantly increased 1, 2, and 5 days after MCAo induction relative to levels in control rats before MCAo induction (0 day vs. 1 day: 3.12 ± 0.82% vs. 22.54 ± 2.44%, *p* < 0.001; 0 day vs. 2 day: 3.12 ± 0.82% vs. 19.67 ± 2.13%, *p* < 0.001; 0 day vs. 5 day; 3.12 ± 0.82% vs. 33.61 ± 4.21%, *p* < 0.001; 1 day vs. 5 day: 22.54 ± 2.44% vs. 33.61 ± 4.21%, *p* < 0.001; and 2 day vs. 5 day: 19.67 ± 2.13% vs. 33.61 ± 4.21%, *p* < 0.001) (Figure 4A). Similarly, we observed elevated CCR2 levels at 1 day after MCAo induction and a marked increase on days 2 and 5 after induction (0 day vs. 1 day: 0.49 ± 0.28% vs. 13.18 ± 2.06%, *p* < 0.001; 0 day vs. 2 day: 0.49 ± 0.28% vs. 30.04 ± 1.89%, *p* < 0.001; 0 day vs. 5 day: 0.49 ± 0.28% vs. 30.79 ± 1.43, *p* < 0.001; 1 day vs. 2 day: 13.18 ± 2.06% vs. 30.04 ± 1.89%, *p* < 0.001; and 1 day vs. 5 day: 13.18 ± 2.06% vs. 30.79 ± 1.43, *p* < 0.001) (Figure 4B). Therefore, we speculated that increased CCR2 levels in the brain following stroke might promote translocation of CCL2-overexpressing hUC-MSCs to the brain parenchyma.

### 2.5. Increased Engraftment of CCL2-Overexpressing hUC-MSCs in Areas of Upregulated CCR2 Levels in the Ischemic Brain

To investigate the increased engraftment of IV-administrated CCL2-overexpressing MSCs in the brain, we immunostained for Stem121, a marker of human neural stem cells, 1 day after the second transplantation (Figure 4C). The results indicated the presence of Stem121+ cells in both the M-MSC and CCL2-MSC groups, whereas Stem121+ cell was not observed the in control and MAN groups. This suggested that IV-transplanted mannitol-treated hUC-MSCs entered the brain parenchyma across the BBB. Interestingly, the number of invaded Stem121+ cells was significantly higher in the CCL2-MSC group relative to that in the M-MSC group (M-MSC vs. CCL2-MSC: 10.39% ± 1.48 vs. 15.8 ± 2.14; *p* < 0.01). Additionally, Stem121+ cells were predominantly detected in infarct and peri-infarct areas but sparsely detected in the contralateral normal hemisphere (data not shown). These results suggested that hUC-MSCs preferentially migrated to the infarct region, and that this activity was promoted by increased CCL2 and/or CCR2 levels.

To analyze the relationship between CCL2 or CCR2 level and CCL2-overexpressing hUC-MSC migration, we co-stained CCL2 or CCR2 and Stem121 on the day 5 after MCAo induction. The results showed no clear correlation between CCL2 expression and the Stem121 distribution (data not shown); however, Stem121+ cells in the CCL2-MSC group co-localized to greater degree with CCR2+ cells or were predominantly found in vicinities exhibiting elevated CCR2 levels as compared with observations in the M-MSC group (M-MSC vs. CCL2-MSC: 36.48 ± 6.06 vs. 64.95 ± 7.64; *p* < 0.01) (Figure 4D). These findings suggested that CCL2-overexpression induced MSC migration into areas exhibiting upregulated CCR2 areas due to CCL2/CCR2 interaction.

### 2.6. CCL2 Expression is Sustained in the Brain Following CCL2-Overexpressing hUC-MSC Transplantation

To evaluate changes in CCL2-expression level according to treatment group, we performed repeated CCL2 immunostaining at five days and four weeks after MCAo induction. The results revealed that CCL2 expression on day 5 was significantly increased in the CCL2-MSC group relative to that in the other groups (control vs. CCL2-MSC: 18.95 ± 3.75% vs. 33.61 ± 4.21%, *p* < 0.001; MAN vs. CCL2-MSC: 17.33 ± 3.63% vs. 33.61 ± 4.21%, *p* < 0.001; and M-MSC vs. CCL2-MSC: 22.93 ± 2.47% vs. 33.61 ± 4.21%, *p* = 0.003) (Figure 4E). Moreover, these differences between groups were sustained until week 4, at which time CCL2 levels remained elevated in the CCL2-MSC group relative to the other groups (control vs. M-MSC: 8.79 ± 0.63% vs. 13.80 ± 1.87%, *p* < 0.01; control vs. CCL2-MSC: 8.79 ± 0.63% vs. 25.79 ± 2.82%, *p* < 0.001; MAN vs. CCL2-MSC: 11.05 ± 1.60% vs. 25.79 ± 2.82%, *p* < 0.001; and M-MSC vs. CCL2-MSC: 13.80 ± 1.87% vs. 25.79 ± 2.82%, *p* < 0.001) (Figure 4E). These results suggested that the increased CCL2 expression following transplantation of CCL2-overexpressing hUC-MSCs was sustained and might affect endogenous brain repair.

### 2.7. Transplantation of CCL2-Overexpressing hUC-MSCs Increases Angiogenesis in the Peri-Infarct Area

Human MSCs promote endogenous brain repair by enhancing angiogenesis in the peri-infarct area following stroke. To evaluate the degree of angiogenesis after transplantation of CCL2-overexpressing hUC-MSCs following brain ischemia, we immunostained for VEGF and RECA-1 to evaluate angiogenesis and microvessel density in the peri-infarct area, respectively. The results indicated a significantly higher number of VEGF+ cells in the M-MSC and CCL2-MSC groups relative to the control group, with the highest number of VEGF+ cells observed in the CCL2-MSC group (control: 15.16 ± 1.13%, MAN: 16.45 ± 1.74%, M-MSC: 19.21 ± 1.55%, and CCL2-MSC: 27.19 ± 1.19%; CCL2-MSC vs. other groups: *p* < 0.001) (Figure 5A). Moreover, RECA-1 immunostaining revealed a significantly higher vessel-coverage area in the CCL2-MSC group relative to then other groups (control: 3.92 ± 1.00%, MAN: 4.42 ± 1.63%, M-MSC: 4.41 ± 0.85%, and CCL2-MSC: 8.15 ± 1.31%; CCL2-MSC vs. control, MAN, and M-MSC: *p* = 0.001, *p* = 0.003, and *p* = 0.003, respectively) (Figure 5B). These findings suggested that CCL2-overexpressing hUC-MSCs enhanced peri-infarct angiogenesis following stroke.

### 2.8. Transplantation of CCL2-Overexpressing hUC-MSCs Increases Endogenous Neurogenesis in the Peri-Infarct Area

To evaluate endogenous neurogenesis in the peri-infarct area, we performed immunostaining for NeuN, BrdU, and DCX. The results showed that the M-MSC and CCL2-MSC groups exhibited significantly higher numbers of NeuN+ cells (control, 16.25 ± 2.20%; MAN, 17.32 ± 2.32% vs. M-MSC, 25.46 ± 0.62% vs. CCL2-MSC, 34.99 ± 2.90%; *p* < 0.001) (Figure 6B), BrdU+ cells (control, 14.77 ± 1.80%; MAN, 16.47 ± 1.21% vs. M-MSC, 23.69 ± 2.04%, vs. CCL2-MSC, 31.17 ± 2.92; *p* < 0.001) (Figure 6C) relative to the MAN and control groups. Additionally, the number of DCX+ cells was higher in the M-MSC and CCL2-MSC groups relative to the control group (control, 13.3 ± 1.20% vs. M-MSC, 23.34 ± 2.45 vs. CCL2-MSC, 32.45 ± 2.73; *p* < 0.001) (Figure 6D). All immunostaining results indicated that the CCL2-MSC groups showed a significantly higher number of positive cells relative to the M-MSC group (*p* < 0.001). These findings suggested that CCL2-overexpressing hUC-MSCs enhanced endogenous neurogenesis in the peri-infarct area following stroke.

### 2.9. Transplantation of CCL2-Overexpressing hUC-MSCs Attenuates Inflammatory Reactions in the Peri-Infarct Area

To investigate the effect of CCL2-overexpressing hUC-MSCs on the inflammatory response following ischemic damage in the brain, we performed immunohistochemical analysis. The inflammatory responses were mainly found around the peri-infarct region in all groups. In GFAP levels, there was no significant difference between the control and MAN groups; however, significant reductions were observed in MSC groups relative to the other groups, with a more significant reduction observed in the CCL2-MSC group relative to the M-MSC group (control, 100 ± 6.5%; MAN, 94.20 ± 4.80; M-MSC, 75.23 ± 1.54; and CCL2-MSC, 55.47 ± 1.65; M-MSC vs. CCL2-MSC: *p* < 0.001) (Figure 7A). Additionally, Iba-1 levels in the vicinity of the infarct area significantly decreased in the MSC groups relative to the other groups, with a more significant reduction observed in the CCL2-MSC group relative to the M-MSC group (M-MSC vs. CCL2-MSC: 51.36 ± 6.13 vs. 34.78 ± 2.28; *p* < 0.05) (Figure 7B).

A previous study reported that in the micro-adherent cell/macrophage phenotype around the infarct area, the M1 macrophage phenotype induces post-ischemic inflammation, whereas the M2 phenotype exhibits neuroprotective effects [14], suggesting that inducing M2 and reducing M1 polarization following ischemic brain injury might promote brain recovery. To confirm changes in the these phenotypes, we stained for iNOS (an M1 marker) and CD206 (an M2 marker) among ED1(CD68)+ cells. First, the results confirmed no changes in the proportion of ED1+ cells around the infarct area in the control and MAN groups (Figure 7C), whereas the M-MSC and CCL2-MSC groups showed a significant decrease in the percentage of ED1+ cells relative to the control and MAN groups (control, 36.10 ± 4.10%; MAN, 36.77 ± 4.40%; M-MSC, 26.90 ± 1.63%; CCL2-MSC, 16.38 ± 3.02%; control vs. M-MSC: *p* = 0.008; control vs. CCL2-MSC: *p* < 0.001; MAN vs. M-MSC: *p* = 0.005; and MAN vs. CCL2-MSC: *p* < 0.001). Specifically, the CCL2-MSC group showed a significant reduction in ED1+ cells as compared with the M-MSC group (M-MSC vs. CCL2-MSC: 26.90 ± 1.63% vs. 16.38 ± 3.02%; *p* = 0.003). Additionally, the proportion of iNOS+ cells among ED1+ cells around the infarct area in the MAN group did not changed relative to that in the control group (control: 65.35 ± 5.09%; and MAN: 64.25 ± 8.18%) (Figure 7D); however, in the MSC groups, the iNOS+:ED1+ ratio slightly decreased relative to the control group, with the CCL2-MSC group showing a more significant reduction in this ratio than the M-MSC group (M-MSC, 52.90 ± 2.4%; CCL2-MSC, 37.93 ± 1.14%; control vs. M-MSC: *p* < 0.05; control vs. CCL2-MSC: *p* < 0.001; and M-MSC vs. CCL2-MSC: *p* < 0.02) (Figure 7D). Furthermore, the proportion of CD206+ cells among ED1+ cells around the infarct area in the MAN group was similar to that in the control group (control: 50.40 ± 5.21%; and MAN: 50.99 ± 1.43%) (Figure 7E), whereas in the MSC groups, the CD206+:ED1+ ratio increased significantly relative to that in the control group, with the CCL2-MSC group showing a more significant increase than that in the M-MSC group (M-MSC vs. CCL2-MSC: 61.07 ± 2.63 vs. 79.58 ± 4.38; *p* < 0.001) (Figure 7E). These results demonstrated that CCL2-overexpressing hUC-MSCs promoted the repair of damaged brain tissue by regulating the inflammatory environment in the infarct area.

## 3. Discussion

These results demonstrated that CCL2-overexpressing hUC-MSCs significantly improved functional recovery and decreased infarct volume relative to naïve hUC-MSCs in an animal stroke model. We observed increased stem cell engraftment by CCL2-overexpressing hUC-MSCs as compared with naïve hUC-MSCs and found that elevated CCR2 levels in the peri-infarction area during acute phase of stroke promoted the migration of CCL2-overexpressing hUC-MSCs into the brain. Additionally, we observed sustained elevations in CCL2 levels during the acute ischemic phase for up to 4 weeks post-induction following transplantation of CCL2-overexpressing hUC-MSCs, as well as increased angiogenesis, neurogenesis, and decreased neuro-inflammation.

Although the M-MSC group showed significant changes in stroke volume and marker levels for various immunohistochemical analyses relative to the control and MAN groups, we did not observed significant differences in behavioral-test outcomes between the three groups. Previous preclinical studies reported that naïve human MSCs of various origins induce functional recovery following stroke [5,8]. In the present study, we did not observed this, possibly attributable to limited engraftment following IV transplantation of stem cells. IV-transplanted human MSCs are trapped in the lung, liver, and spleen, resulting in low proportions of cells reaching brain parenchyma [15,16]. Therefore, the previous studies described the therapeutic mechanism of IV-transplanted human MSCs as being induced through the bystander effect of remote, trapped MSCs secreting cytokines and trophic factors in extra-brain organs [15].

CCL2 is a pro-inflammatory chemokine that plays an important role in inflammatory reactions under various neurological conditions, such as stroke, Alzheimer’s disease, and multiple sclerosis [17]. CCL2 is constitutively present in the normal brain and within neurons, astrocytes, and endothelial cells; however, it is upregulated during a stroke [17,18,19]. Increases in CCL2 level induce the recruitment of various inflammatory cells to the brain and disruption of the BBB after stroke [10,20]. This migration of inflammatory cells is promoted by CCL2/CCR2 interactions, which induces translocation of CCR2-expressing inflammatory cells into infarct areas already experiencing increased CCL2 expression following a stroke. Previously studies suggest that increased CCL2 levels in stroke exacerbate neuro-inflammation and subsequent brain injury, thereby increasing stroke volume and resulting in poor prognosis [21,22]. However, recent studies show that CCL2/CCR2 interaction positively affects functional recovery from stroke [23,24], and other studies confirm that CCL2 promotes the homing of CCR2-expressing stem cells from the bone marrow to the damaged brain, which contributes to brain repair following stroke [20,25,26]. Additionally, this interaction promotes intravascular migration of stem cells into the injury sites and is essential for the therapeutic homing of CCR2-expressing stem cells [6,7,11,27,28,29]. Furthermore, studies indicate that CCR2 overexpression on MSCs [20] or exosomes [26] following genetic modification enhances their targeted migration to the ischemic hemisphere and improves therapeutic outcomes.

In addition to CCR2 presentation on inflammatory cells, such as monocytes/macrophages and lymphocytes, it is also found in neurons and astrocytes [30], with reports showing increased CCR2 levels in the brain following ischemia [31] and heat stroke [32]. Therefore, we speculated that the CCL2/CCR2 interaction might promote the migration of CCL2-overexpressing hUC-MSCs into areas exhibiting elevated CCR2 levels. Our results confirmed that both CCR2 and CCL2 levels increased in brain parenchyma during acute phase after stroke, and that CCL2-overexpressing hUC-MSCs more effectively entered the brain parenchyma relative to naïve hUC-MSCs and distributed more densely around areas of upregulated CCR2. This suggests that increased engraftment of CCL2-overexpressing hUC-MSCs effectively induced cell-to-cell communication with host cells to promote the secretion of cytokines and trophic factors to nearby damaged brain areas and neurovascular niche structures. Our findings showed that the CCL2-MSC group underwent significant functional recovery and displayed decreased stroke volume relative to the M-MSC group.

Previous studies using animal models of stroke report that MSCs increase angiogenesis and neurogenesis [3,5,33]. In the present study, both MSC groups showed increased endogenous angiogenesis (VEGF+ cells and RECA-1+ areas) and neurogenesis (NeuN+, BrdU+, and DCX+ cells); however, we found that these levels were enhances in the CCL2-MSC group relative to the M-MSC group. It is possible that this is a consequence of the increased number of migrated stem cells to the infarct area. A recent report indicated that CCL2 participates in mechanisms associated with neurologic recovery [12]. In the present study, we found sustained elevations of CCL2 levels in the brain at 28 days after stroke induction in the CCL2-MSC group as compared with levels in the other groups. This suggests that this activity might promote increased functional recovery.

Previous studies reported that CCL2 is implicated in angiogenesis and promotes capillary-like structure formation of human umbilical vein endothelial cells in vitro through the increased expression of both VEGF and hypoxia-inducible factor 1-alpha, and through activation of the Ets-1 transcription factor [34,35]. Additionally, secreted CCL2 mediates the angiogenic effect of tissue factors by recruiting smooth muscle cells toward endothelial cells and facilitates the maturation of newly formed microvessels in Matrigel plugs in vivo [34,36]. Moreover, the CCL2/CCR2 axis plays a critical biological role in recovery of blood flow in a murine hindlimb ischemic model and might have a regulatory role in both the migration of endothelial cells and maturation of neovascularization [34].

Several studies indicate that CCL2 has beneficial neuroprotective effects beyond its established role in leukocyte recruitment and activation [12,30,37]. Previous in vitro studies showed that in mixed cultures of human or mouse neurons and astrocytes, CCL2 provided neuroprotection against apoptotic stimuli induced by N-methyl-D-aspartate (NMDA) and Tat protein. CCL2 treatment inhibits the increase of toxic extracellular glutamate concentration induced by Tat or NMDA and regulates the intracellular trafficking of Tat and NMDA receptor 1 expression [12,38]. Additionally, reports indicate that norepinephrine and adenosine increase CCL2 levels, resulting in reductions of neuronal damage attributable to NMDA or glutamate [12,39]. Furthermore, hypoxic preconditioning induces stroke tolerance in mice via CCL2 signaling pathway [37,40].

The exact role of CCL2 in normal brain function or neurodevelopment remains unclear; however, studies suggest possible roles in adult neurogenesis and/or repair processes through the CCR2 on neural progenitor cells [30]. It has been reported that CCL2 overproduced by activated astrocytes and microglia in an MCAo model induced the migration of newly formed CCR2-expressing subventricular zone neuroblasts from neurogenic regions to the damaged regions of brain following focal ischemia [12,41,42]. In the present study, continuous increases in CCL2 levels in the brain following transplantation of CCL2-overexpressing hUC-MSCs might mediate the various steps of neurogenesis, such as neuronal progenitor proliferation, neuroblast migration toward the injured area, and differentiation, maturation, and integration of newly generated neurons [12,41,42].

Furthermore, in our study, the sustained increase in CCL2 levels in the brain following transplantation of CCL2-overexpressing hUC-MSCs might induce further increases in the migration of monocytes and macrophages, which are involved in neurological recovery after stroke. Microglia/macrophages are the primary immune cells involved in defending against brain damage. After a stroke, excessive activation by damage-related mechanisms can destroy nerve cells and the BBB, thereby negatively affecting neuronal development. However, recent studies suggest that activated microglia contribute to neural plasticity and neuro-restoration following insult [41]. In response to various signals, microglia/macrophages polarize to two major phenotypes: pro-inflammatory (M1) and anti-inflammatory (M2) [41,43]. Although the M1 population of microglia/macrophages mainly exhibits destructive properties, the M2 population plays a protective role in nerves; therefore, regulating macrophage polarization can be an important factor in improving stroke [14]. MSCs reportedly weaken the M1 population and induce activation of M2 polarization [44,45]. In the present study, the results showed that transplantation of CCL2-overexpressing hUC-MSCs decreased neuro-inflammation accompanied by a decreased M1 population and increased M2 population relative to those in other groups, especially naïve hUC-MSCs. Therefore, the sustained upregulation of CCL2 for up to 28 days of the post-stroke delayed phase following CCL2-overexpressing hUC-MSCs transplantation might contribute to neurological recovery via CCL2/CCR2-dependent microglia/macrophage status based on the dominant pro-inflammatory phenotype observed in the acute phase versus the dominant anti-inflammatory phenotype in delayed phase after stroke [24,46].

In conclusion, we demonstrated that CCL2-overexpressing hUC-MSCs effectively restored functional deficits in an animal stroke model by promoting continuous increases in CCL2 levels in the brain, enhancing angiogenesis and neurogenesis, and decreasing neuro-inflammation. This activity was a result of the increased efficiency of hUC-MSC migration due to elevated CCR2 levels in the infarcted areas of the brain, which promoted migration of the CCL2-overexpressing hUC-MSCs. These findings suggest that genetically modified CCL2-overexpressing MSCs might represent an effective strategy for cell therapy of clinical stroke.

## 4. Materials and Methods

### 4.1. Ethics

All experimental animals were manipulated in accordance with guidelines provided by the Institutional Animal Care and Use Committee of CHA University (IACUC nos. 190088 (1 April 2019–1 April 2020) and 200062 (1 April 2020–1 April 2021).

### 4.2. Culture of hUC-MSCs

hUC-MSCs were provided from CHA Biotech (Seongnam, Korea). The detailed description of preparation and characterization for hUC-MSCs were reported previously [47,48,49]. hUC-MSCs were cultured in high-glucose minimum essential medium (MEM; Gibco, Gaithersburg, MD, USA) supplemented with 10% fetal bovine serum (Gibco, Gaithersburg, MD, USA), 50 µg/mL gentamycin (Sigma-Aldrich, St. Louis, MO, USA), 1 µg/mL heparin (Sigma-Aldrich, St. Louis, MO, USA) and 25 ng/mL fibroblast growth factor-4 (Peprotech, Rocky Hill, NJ, USA). We used cells at passage eight or nine for the experiments.

### 4.3. Manufacturing of CCL2-Overexpressing hUC-MSCs

The pUCIDT-kan plasmid (2705bp) harboring CCL2 was purchased from Cosmo Genetech (Seoul, Korea). The plasmid also harbored green fluorescent protein (GFP), which allowed confirmation of transfection and CCL2 expression. The CCL2 plasmid was amplified in *Escherichia coli* DH5α cells and purified using a NucleoBond Xtra Midi Plus kit (Macherey-Nagel, Düren, Germany) according to manufacturer instructions. hUC-MSCs (2 × 10^6^) were plated in T-75 flasks, and after a 24-h incubation at 37 °C in a CO_2_ incubator, Lipofectamine–stem-cell-mediated transfection was performed using reagents supplied by Invitrogen (Carlsbad, CA, USA) and according to their instructions. Prior to transfection, the cells were washed twice with phosphate-buffered saline, and the culture medium was changed to serum-free α-MEM. The first dilution of CCL2 plasmids was performed in 12-mL microtubes with 725 μL Opti-MEM (Cat. No. 31985; Gibco, Gaithersburg, MD, USA). In a separate tube, Lipofectamine was added to 725 μL Opti-MEM, followed by transfer of the plasmid mixture and incubation for 10 min at 20 °C to form a DNA-Lipofectamine–stem cell complex. The complex was added to hUC-MSCs, and the mixture was incubated at 37 °C in a CO_2_ incubator. After 6 h, the medium was supplemented with 10% serum, and after another 24 h, GFP expression was observed by fluorescence microscopy (LSM510; Carl Zeiss Microimaging Inc., München, Germany; Nikon Eclipse Ni; Nikon Instruments Inc., Melville, NY, USA), indicating successful transfection.

### 4.4. Reverse Transcription Polymerase Chain Reaction (RT-PCR)

RT-PCR was performed to evaluate CCL2 mRNA levels in CCL2-overexpressing hUC-MSCs. Total RNA was extracted from hUC-MSCs using TRIzol reagent (Ambion, Austin, TX, USA). After cDNA synthesis, specific primers for CCL2 (F: 5′-TTTGGTTGCATGAAGGCTGC-3′; and R: 5′-GCCGAACTTTCTGGTCCTCA-3′) and *glyceraldehyde 3-phosphate dehydrogenase* (*GAPDH*; F: 5′-AGCAATGCCTCCTGCACCACCAAC-3′; and R: 5′-CCGGAGGGGCCATCCACAGTC-3′) were used for amplification using the Quantitect SYBR Green PCR kit (Qiagen, Hilden, Germany) according to the following program: 40 cycles at 95 °C for 15 s and 60 °C for 60 s. CCL2 levels were normalized against those of *GAPDH*, and amplifications were independently replicated four times on different days.

### 4.5. Western Blot

Western blot was used to evaluate CCL2 levels in CCL2-overexpressing hUC-MSCs. After hUC-MSCs were harvested and lysed in 2× Laemmli sample buffer (Bio-Rad, Hercules, CA, USA), the proteins were isolated. The proteins obtained were quantified using the BCA Assay kit (Thermo Fisher Scientific, Waltham, MA, USA). And then, 50 µg proteins were separated by 10% sodium dodecyl sulfate polyacrylamide gel electrophoresis and transfer to polyvinylidene difluoride membranes (Millipore, Billerica, MA, USA) using standard electroblotting procedures. The blots were blocked with 5% skim milk in Tris-buffered saline containing Tween-20 for 1 h at room temperature and immunolabeled with primary antibodies against CCL2 (1:1000; Abcam, Cambridge, UK) and β-actin (1:1000; Santa Cruz Biotechnology, Dallas, TX, USA) overnight at 4 °C. Immunolabeling was detected with an enhanced chemiluminescence kit (Bio-Rad, Hercules, CA, USA) using the LAS4000 imaging system (GE Healthcare, Pittsburgh, PA, USA). Experiments were independently replicated four times on different days.

### 4.6. MCAo Stroke Model

Male Sprague–Dawley rats (7-weeks old; 270–300 g) were purchased from Orient Bio (Orient Bio, Seongnam, Korea) and housed in a temperature- and humidity controlled room with a 12-/12-h light/dark cycle. Animals were allowed to acclimatize to the laboratory 1-week prior to surgery and had ad libitum access to food and water. Rats were anesthetized via intramuscular injection of ketamine and Rompun (3:1), followed by placement on a heating pad, limb fixation with tape, and monitoring of body temperature. The internal, external, and common carotid arteries (ECAs and CCAs, respectively) were exposed after midline skin incision in the right neck area, followed by ligation of CCA branches. After incision of the ECA, a filament coated with silicon (Doccol Corp., Sharon, MA, USA) was inserted 20 mm from the ECA through the carotid artery to block the MCA. After 60 min of occlusion, blood flow was restored in the CCA, and the MCA was reperfused by withdrawing filament. During the procedure, rectal temperature was maintained at 37 °C by heating surgical pad. The following day, the MCAo-induced rats were randomly divided into four groups (*n* = 10/group), (1) phosphate-buffered saline (PBS) group (control), (2) mannitol-treatment group (MAN), (3) mannitol+hUC-MSC-treatment group (M-MSC), and (4) mannitol+CCL2-overexpressing hUC-MSC-treatment group (CCL2-MSC).

### 4.7. Stem Cell Transplantation

The in vivo experiments are described in Figure 8. Mannitol, which is effective at increasing the permeability of the blood–brain barrier (BBB) [50], was injected prior to implantation. According to cell administration described in a previous study [47], intravenous (IV) mannitol (2.5 mL of 20% solution; injection speed: 1.0 mL/min) and either IV hUC-MSCs or IV CCL2-overexpressing hUC-MSCs (1 × 10^6^ cells in 0.5 mL PBS) were carefully injected into the tail vein at two different transplantation time points (1- and 4-days post-MCAo induction) in the cell-transplantation group. Immunosuppressive drugs were not used. The PBS group received only IV PBS, and the MAN group received only IV mannitol without cell transplantation at the same time points. During the procedure, no profound bleeding occurred during transplantation, and vital signs in rats were stable. There were no perioperative complications (death or severe morbidity) in any of the four tested groups or significant changes in temperature and weight (data not shown). All rats, including those receiving CCL2-overexpressing hUC-MSC transplantation, survived during the experiment, supporting the safety of these hUC-MSCs.

### 4.8. Behavioral Tests

Behavioral tests were performed independently by two blinded investigators at 1-, 2-, and 5-days after MCAo induction, followed by once weekly for up to four weeks. The mNSS test assessed scores according to analysis of motor, sensory, reflex, and balance deficits and ranged from 0 to 28. Higher scores indicate closer to severe conditions, whereas lower scores indicate normal conditions. Stepping tests to assess forefoot deficits were performed using a treadmill. Recovery of forefoot motor function was measured by counting the number of hand strokes on the bottom of the treadmill. For each forefoot test, the average of three times trials was used for analysis. Foot-fault tests measured forefoot motor impairment using a ladder (120-cm long with rungs 4-cm apart). The falling depth of the right forelimb was counted when the forelimb of the affected side slid, fell off of, or slipped between the rod, and scores were recorded as follows: fallen to the shoulder: 2; fallen to the ankle: 1; and normal walk: 0 [47].

### 4.9. Measurement of Infarct Size

At 4-weeks post-MCAo induction, infarct volume was measured in the independent MCAo model group using 2,3,5-triphenyltetrazolium chloride (TTC) and Cresyl Violet staining. The removed brain was cut at 1-mm intervals and stained with 2% TTC solution (*n* = 5/group), followed by fixation with 4% paraformaldehyde (PFA). The infarct volume of the MCAo model was also measured using Cresyl Violet staining of brain sections in each group. The infarct size was measured as a percentage of the intact hemisphere using the following formula: Infarct size (%) = [1 − (area of remaining ipsilateral hemisphere/area of intact contralateral hemisphere)] × 100. The size of the infarct region was measured with Image J software (National Institutes of Health, Bethesda, MD, USA). Percentages of infarct size were summed from three sections per brain, and values obtained from TTC and Cresyl Violet staining were compared.

### 4.10. Immunohistochemistry

Rats were euthanized and transcardially perfused with heparinized saline and 4% PFA (Merck Millipore, Waltham, MA, USA). After perfusion, the brain was separated and fixed with 4% PFA overnight at 4 °C and then transferred to 30% sucrose solution (Sigma-Aldrich, St. Louis, MO, USA) for incubation for at least 3 days at 4 °C. The brains were then embedded using optimal cutting temperature compound and stored at −80 °C. Brain tissues were then sectioned at a thickness of 40 µM on a cryostat. Free-floating sections were blocked with 5% normal goat serum in PBS for 1 h at 20 °C and incubated with blocking buffer containing the following primary antibodies overnight at 4 °C: anti-Stem121 (1:200; Takara Bio, Shiga, Japan), anti-neuronal nuclei (NeuN; 1:200, Merck Millipore, Waltham, MA, USA), anti-bromodeoxyuridine (BrdU; 1:200; Invitrogen, Carlsbad, CA, USA), anti-doublecortin (DCX; 1:200, Cell Signaling Technology, Danvers, MA, USA), anti-vascular endothelial growth factor (VEGF; 1:200; Thermo Fisher Scientific, Waltham, MA, USA), anti-rat endothelial cell antigen-1 (RECA-1; 1:100; Abcam, Cambridge, UK), anti-ionized calcium-binding adapter molecule-1 (Iba-1; 1:500; Wako Pure Chemical Co., Kyoto, Japan), anti-glial fibrillary acidic protein (GFAP; 1:200; BD Pharmingen, San Diego, CA, USA), anti-ED1/CD68 (1:500; Bio-Rad, Hercules, CA, USA), anti-inducible nitric oxide synthase (iNOS; 1:50; Thermo Fisher Scientific, Waltham, MA, USA), anti-CD206 (1:100; Thermo Fisher Scientific, Waltham, MA, USA), anti-CCL2 (1:200; Abcam, Cambridge, UK), and anti-CCR2 (1:250; Abcam, Cambridge, UK). The sections were washed in PBS three times for 15 min each and then incubated with secondary antibodies (1;500, Invitrogen, Carlsbad, CA, USA) for 1 h at 20 °C. The sections were then treated with the nuclear marker 4′,6-diamidine-2′-phenylindole dihydrochloride (Sigma-Aldrich, St. Louis, MO, USA). The sections were cover-slipped with mounting solution (Vector Labs, Burlingame, CA, USA) to delay fading. Changes in CCL2 and CCR2 levels in the brain during acute ischemia were confirmed on the day before MCAo induction and days 1, 2 and 5 after MCAo induction. To investigate the migration and engraftment of IV-transplanted cells in the brain, Stem121 analysis was performed 1 day after the second transplantation. Co-localization of Stem121 and, CCL2 or CCR2 was also performed 1 day after the second transplantation. Changes in CCL2 level according treatment were determined at 5 days and 4 weeks after MCAo induction. Other immunohistochemical analyses to assess angiogenesis, neurogenesis, and neuro-inflammation were performed 4 weeks after MCAo induction. All immunohistochemical analyses were performed four or five times for each group.

### 4.11. Quantitation of Vessel Density

Vascular images detected by RECA-1 targeting to identify the vascular endothelium were transformed into a black-and-white image using VesSeg-Tool software (http://www.isip.uni-luebeck.de/index.php?id=150&L=0, Lübeck, Germany), with RECA-1+ areas shown in white calculated as a percentage of the total area.

### 4.12. Statistical Analysis

Statistical analyses were performed using GraphPad Prism software (v.5.0; GraphPad Software, La Jolla, CA, USA) and SigmaPlot software (v.14.0.; https://systatsoftware.com/products/sigmaplot/, San Jose, CA, USA). Statistically significant differences between two groups were analyzed using Student’s *t* test, and analysis of behavioral test results was performed using one-way analysis of variance (ANOVA). Statistically significant differences in multiple comparisons of immunohistochemical results were analyzed using one-way ANOVA, followed by Tukey’s post hoc test. A *p* < 0.05 was considered significant, and all values are presented as the mean ± standard error (SEM).

## Figures and Tables

**Figure 1 ijms-21-07795-f001:**
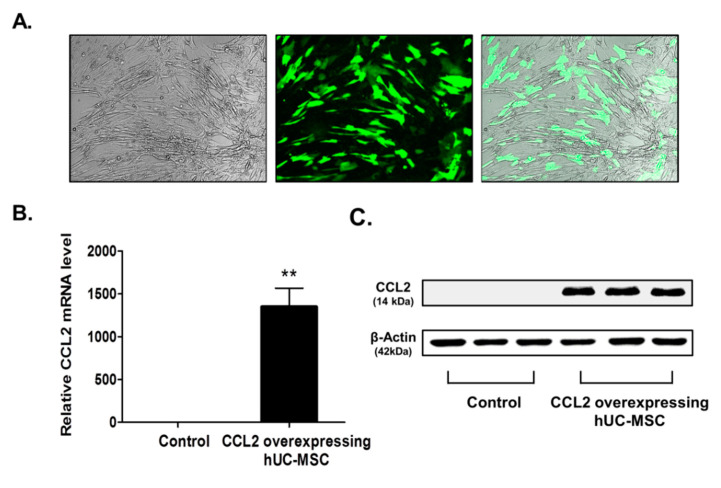
Confirmation of CCL2 overexpression following transfection into hUC-MSCs. (**A**) At 24-h post-transfection, hUC-MSCs observed in the bright field expressed GFP by fluorescence microscopy, and (**B**) RT-PCR results indicated CCL2 mRNA levels. Data represent fold changes relative to GAPDH and are presented as the mean ± SEM. ** *p* < 0.01, Student’s *t* test. (**C**) Western blot analysis showing CCL2 protein levels at 24-h post-transfection. CCL2-specific bands were quantified and normalized against those of β-actin. Scale bar = 100 µm.

**Figure 2 ijms-21-07795-f002:**
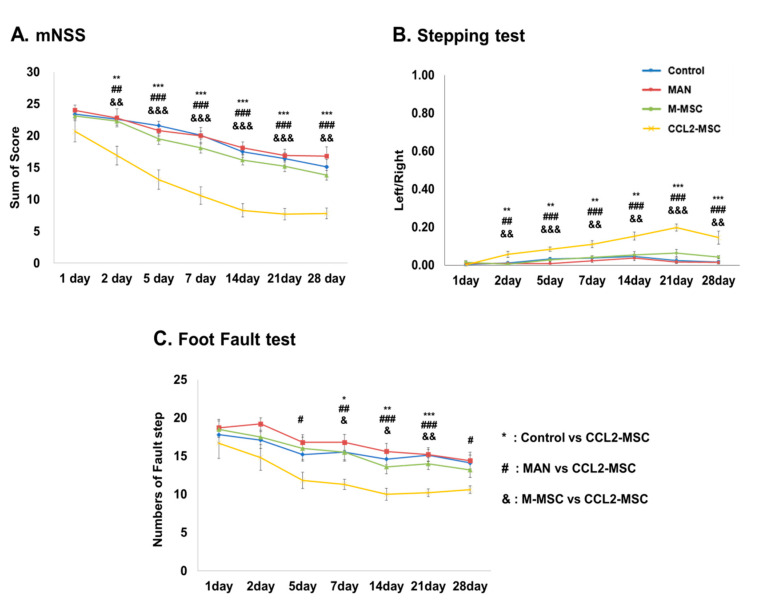
Evaluation of hUC-MSC transplantation according to behavioral testing. Changes in (**A**) mNSS, (**B**) stepping, and (**C**) foot-fault results for each group for up to 28 days after MCAo induction. Data represent the mean ± SEM and were analyzed by one-way ANOVA. * *p* < 0.05, ** *p* < 0.01, *** *p* < 0.001, control vs. CCL2-MSC; # *p* < 0.05, ## *p* < 0.01, ### *p* < 0.001, MAN vs. CCL2-MSC; & *p* < 0.05, && *p* < 0.01, &&& *p* < 0.001, M-MSC vs. CCL2-MSC.

**Figure 3 ijms-21-07795-f003:**
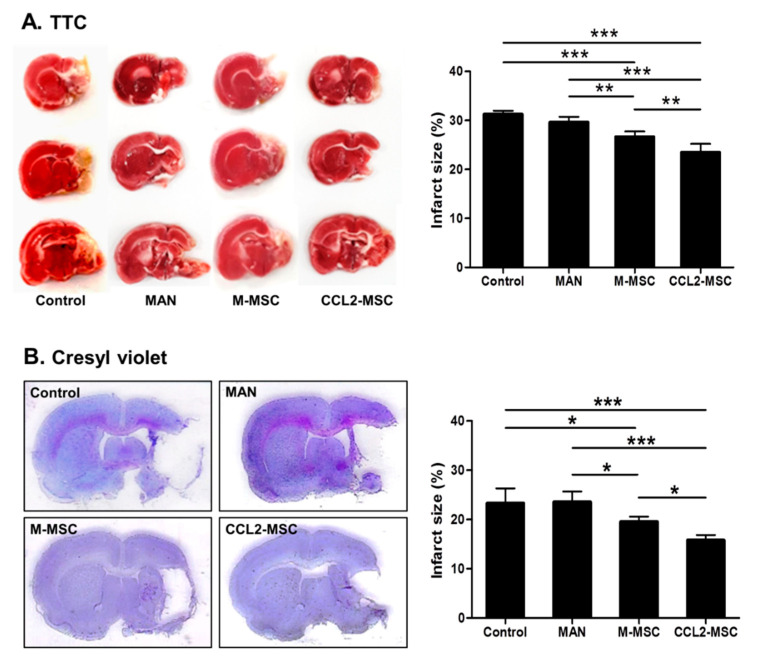
Comparison of infarct size between treatment groups. Infarct size was measured by (**A**) TTC and (**B**) Cresyl Violet staining at 4 weeks after MCAo induction. Data represent the mean ± SEM and were analyzed by one-way ANOVA. * *p* < 0.05, ** *p* < 0.01, *** *p* < 0.001.

**Figure 4 ijms-21-07795-f004:**
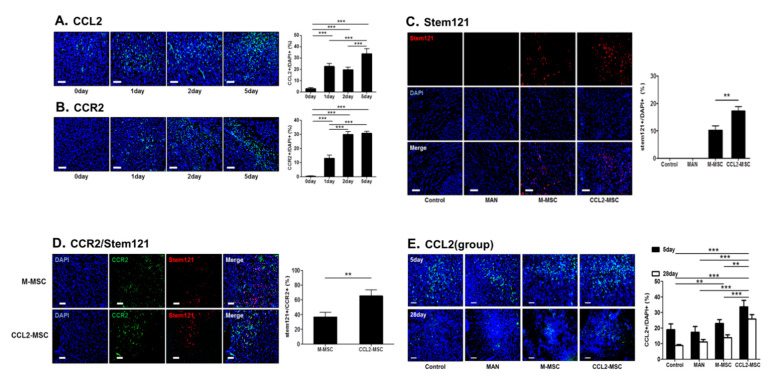
Changes in CCL2 and CCR2 levels in the brain during the acute stroke phase and correlations between MSC migration and CCR2 levels following transplantation. (**A**,**B**) Representative immunostaining images showing changes in CCL2 and CCR2 levels on days 1, 2 and 5 after MCAo induction as compared before induction. The graph shows the number of CCL2+ or CCR2+ cells in each group. Data were analyzed by one-way ANOVA test. (**C**) The CCL2-MSC group showed enhanced engraftment into the peri-infarct area relative to the M-MSC group. The graph shows the number of Stem121+ cells in the peri-infarct area in each group. Data were analyzed using a paired Student’s *t* test. (**D**) Co-immunostaining images of CCR2+ and Stem121+ cells in the peri-infarct area in each group on 5 day after MCAo induction. The graph shows the number of Stem121+/CCR2+ cells in each group. Data were analyzed using a paired Student’s *t* test. (**E**) Representative immunostaining images showing changes in CCL2 levels at 5 and 28 days after MCAo induction between treatment groups. The graphs show the number of CCL2+ cells in each group. Data represent the mean ± SEM and were analyzed by one-way ANOVA. ** *p* < 0.01, *** *p* < 0.001. Scale bar = 100 μm.

**Figure 5 ijms-21-07795-f005:**
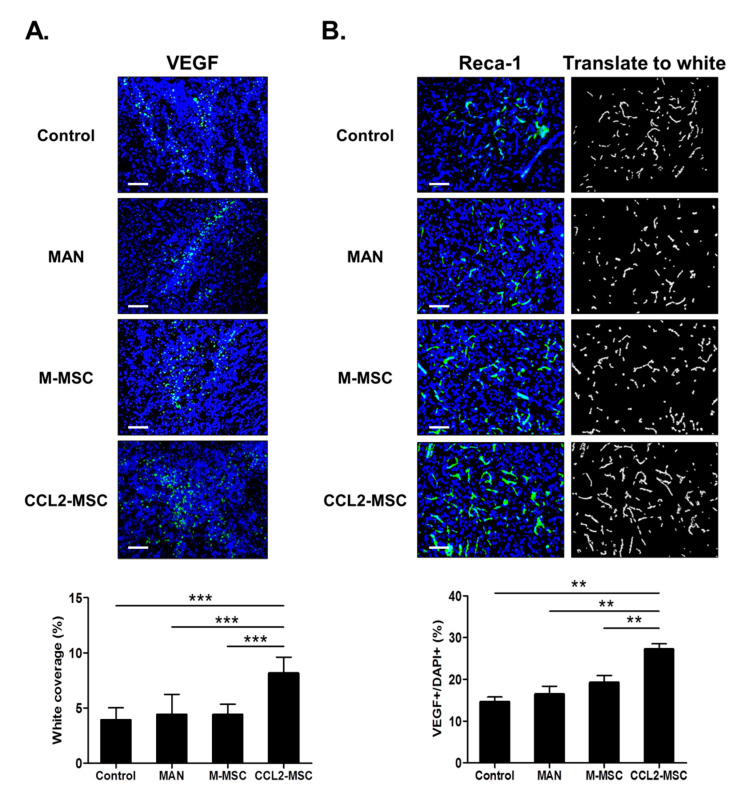
Changes in angiogenesis in the peri-infarct area following hUC-MSC transplantation. Immunostaining analyses were performed 28 days after MCAo induction. (**A**) Representative immunostaining images showing VEGF+ cells in each group. The graph shows the number of VEGF+ cells in each group. (**B**) Representative immunostaining image showing RECA-1+ vessels in each group. The graph shows vessel density in the peri-infarct area in each group. Data represent the mean ± SEM (*n* = 5/group) and were analyzed by one-way ANOVA. ** *p* < 0.01, *** *p* < 0.001. Scale bar = 100 μm.

**Figure 6 ijms-21-07795-f006:**
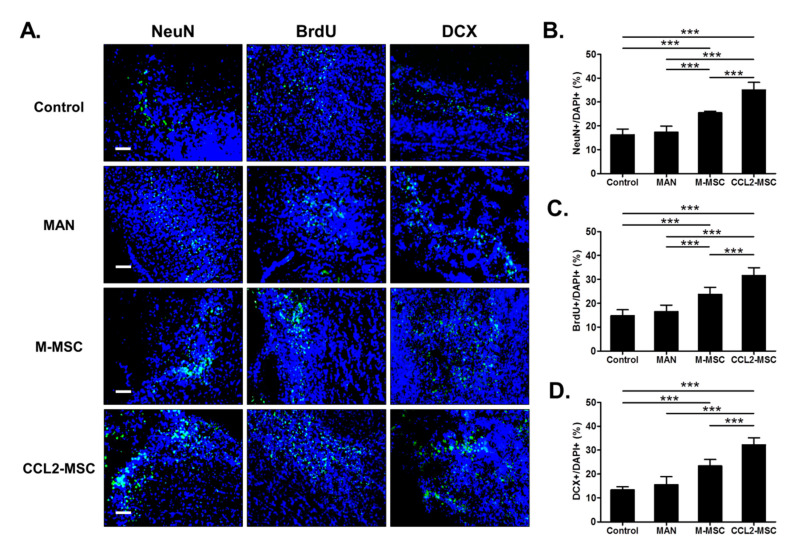
Changes in endogenous neurogenesis in the peri-infarct area following hUC-MSC transplantation. (**A**) Representative immunostaining image showing the distribution of NeuN+, BrdU+, and DCX+ cells at 28 days after MCAo induction in each group. The graphs show the numbers of (**B**) NeuN+, (**C**) BrdU+, and (**D**) DCX+ cells in each group. Data represent the mean ± SEM and were analyzed using one-way ANOVA. *** *p* < 0.001). Scale bar = 100 µm.

**Figure 7 ijms-21-07795-f007:**
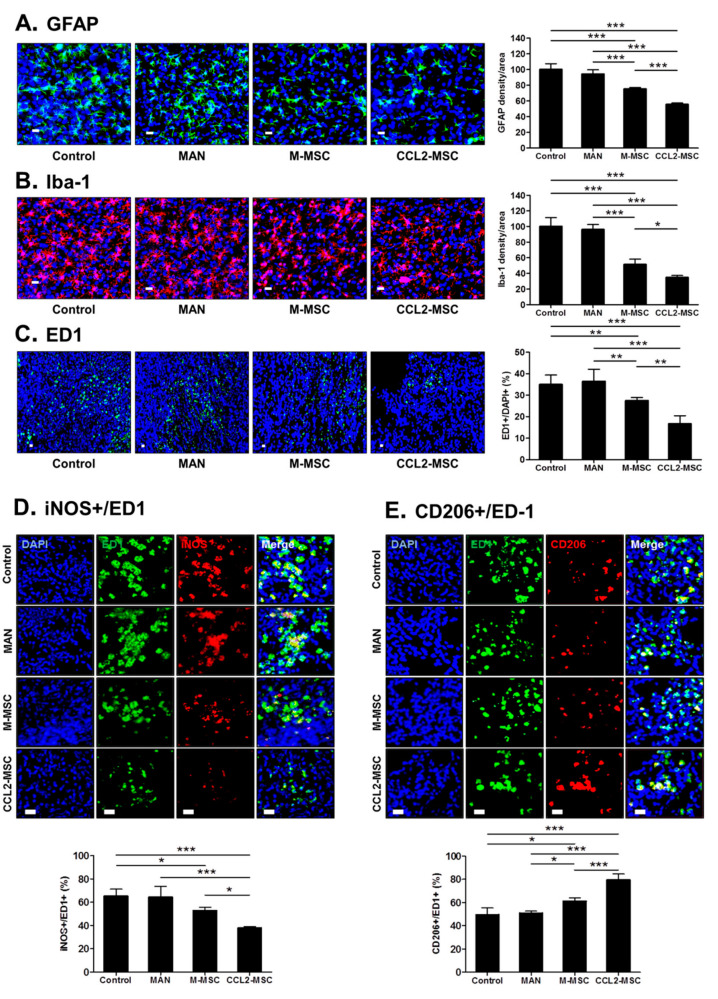
Changes in inflammation in the peri-infarct area following hUC-MSC transplantation. Immunostaining analyses were performed 28 days after MCAo induction. (**A**) Images showing GFAP+ cells in the peri-infarct area. The graph shows the density of GFAP+ cells per area in each group. (**B**) Representative immunostaining images showing Iba-1+ cells in the peri-infarct area. The graph shows Iba-1+ cells per area in each group. (**C**) Representative immunostaining images showing ED-1+ cells in the peri-infarct area. The graph shows the number of ED1+ cells in each group. (**D**, **E**) Immunostaining images showing iNOS+ cells and CD206+ cells in the peri-infarct area in each group. Each graph shows the iNOS+:ED1+ cell ratio and CD206+:ED-1+ cell ratio in each group. Data represent the mean ± SEM and were analyzed by one-way ANOVA. * *p* < 0.05, ** *p* < 0.01, *** *p* < 0.001. Scale bar = 20 μm.

**Figure 8 ijms-21-07795-f008:**
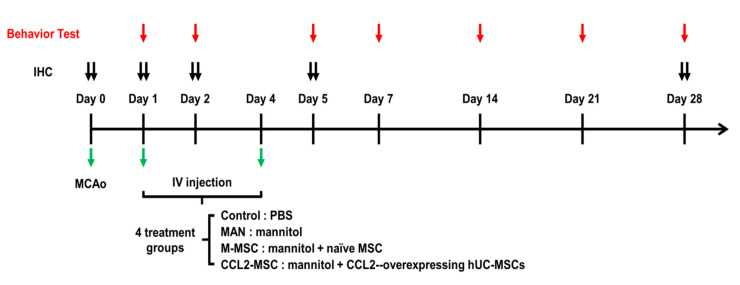
Experimental schedule. Schematic showing the timing of cell transplantation and various analyses in the MCAo animal model, including dates of transplantation, behavior test, and immunohistochemical analyses. IHC: immunohistochemical assay.

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
