# Peer review of "Enhancing the Therapeutic Potential of *CCL*2-Overexpressing Mesenchymal Stem Cells in Acute Stroke"

_ijms, 2020, doi:10.3390/ijms21207795_

Round 1
Reviewer 1 Report
This is a very interesting study that advances the therapeutic potential of CCL2 overexpressing MSCs in the treatment of stroke.
The background, methods and discussion are all well presented. There are however, some concerns with the results section. Namely, the size of the graphs and images needs to be enlarged for clarity.
Regarding presentation of P values. Typically, P= is limited to values = or > than 0.5; whereas significant values are denoted with < (for instance P=0.0019 should be reported as P< 0.002)
Author Response
Dear Reviewer,
I greatly appreciate you for the thorough review of the paper with valuable comments. The following are the answers to your comments.
Point 1: The size of the graphs and images needs to be enlarged for clarity.
Response 1: We totally agree that more clarity is needed. We have revised the graphs and images to make clear.
Point 2: Regarding presentation of P values. Typically, P= is limited to values = or > than 0.5; whereas significant values are denoted with < (for instance P=0.0019 should be reported as P< 0.002)
Response 2: We agree that there is a risk of confusion. So, we have revised the paragraph to make clear as the reviewer's valuable comment.
Again, thank you very much for the valuable comments. They have helped us produce a more enriched and refined paper. Because we know the reputation of International Journal of Molecular Sciences, it would be a great honor to be able to publish our paper. Thank you again for your valuable comments and favorable review.

Reviewer 2 Report
In the present work, Lee et al., investigated the therapeutic potential of CCL2- 3 overexpressing mesenchymal stem cells in acute stroke. In particular, CCL2-overexpressing hUC-MSCs were intravenously transplanted with mannitol in rats with middle cerebral arterial occlusion. These findings indicated that CCL2-overexpressing hUC-MSCs showed better functional recovery compared to naïve hUC-MSCs according to the increased migration of these cells into brain areas of higher CCR2 expression, thereby promoting subsequent endogenous brain repair. The experimental design and idea are adequate and original, however there are some points that required the Authors’ attention:
- Figure 1 A shows bright field and immunofluorescence for GFP. The Bright field is not mentioned in the legends. More importantly the Authors should show a better resolution images to appreciate the merge, since in this form is not possible.
- Figure 1 C shows Western blotting representative figures but are overexposed and saturated. Also, the uncropped figure should be added in the supplementary. It would be better to add also the MW of the protein detected.
- The Authors should check thoroughly the abbreviations, i.e. hUC-MSCs is never spelled out in the main text.
- In the Methods section relative to Western Blotting the amount of protein loaded should be added and the assay used to evaluate the protein content as well (i.e. BCA test).
- Please correct the word “behavior test” in behavioral test in the timeline (Figure 8)
Author Response
Dear Reviewer,
I greatly appreciate you for the thorough review of the paper with valuable comments. The following are the answers to your comments.
Point 1: Figure 1 A shows bright field and immunofluorescence for GFP. The Bright field is not mentioned in the legends. More importantly the Authors should show a better resolution images to appreciate the merge, since in this form is not possible.
Response 1: We agree with you that we did not mention bright field in the legends. So, we mentioned as followings in Figure 1. legend.
In page 3, from line 74 to line 76
“Figure 1. (A) At 24-h post-transfection, hUC-MSCs observed in the bright field expressed GFP by fluorescence microscopy.”
Also, we changed to the better resolution images to appreciate the merge as the reviewer's valuable comment.
Point 2: Figure 1 C shows Western blotting representative figures but are overexposed and saturated. Also, the uncropped figure should be added in the supplementary. It would be better to add also the MW of the protein detected.
Response 2: We agree with you that our figure needs to be modified. So we revised the figure and presented it again as the reviewer's valuable comment. And, we added supplementary figure 1 for the uncropped figures in page 20, from line 693 to line 698
Point 3: The Authors should check thoroughly the abbreviations, i.e. hUC-MSCs is never spelled out in the main text.
Response 3: We agree with you that we should check the abbreviations. So, we have revised the abstract and paragraph to make clear as the reviewer's valuable comment. From: Page 1, line 17, beginning with the words "human umbilical cord-derived..." To: Page 1, line 18, ending with the words "... mesenchymal stem cell) (hUC-MSCs).” And, from: Page 2, line 63, beginning with the words "human umbilical cord-derived..." To: Page 2, line 64, ending with the words "... mesenchymal stem cell) (hUC-MSCs).”
Point 4: In the Methods section relative to Western Blotting the amount of protein loaded should be added and the assay used to evaluate the protein content as well (i.e. BCA test).
Response 4: We agree with the reviewer's valuable comment. So, we have revised the paragraph to make clear as the reviewer's valuable comment. From: Page 13, line 422, beginning with the words "After hUC-MSCs was harvested and lysed with 2X Laemmli sample buffer …” To: Page 13, line 427, ending with the words "... polyvinylidene difluoride membranes (Millipore, Billerica, MA, USA) using standard electroblotting procedures.”
Point 5: Please correct the word “behavior test” in behavioral test in the timeline (Figure 8)
Response 5: We agree that there is a risk of confusion. And, we have revised the figure legends to make clear as the reviewer's valuable comment.
Again, thank you very much for the valuable comments. They have helped us produce a more enriched and refined paper. Because we know the reputation of International Journal of Molecular Sciences, it would be a great honor to be able to publish our paper. Thank you again for your valuable comments and favorable review.
